# Comparative Analysis of Hepatic Gene Expression Profiles in Murine and Porcine Sepsis Models

**DOI:** 10.3390/ijms252011079

**Published:** 2024-10-15

**Authors:** Fëllanza Halimi, Tineke Vanderhaeghen, Steven Timmermans, Siska Croubels, Claude Libert, Jolien Vandewalle

**Affiliations:** 1Department of Pathobiology, Pharmacology and Zoological Medicine, Faculty of Veterinary Medicine, Ghent University, 9820 Merelbeke, Belgium; fellanza.halimi@ugent.be (F.H.); siska.croubels@ugent.be (S.C.); 2VIB Center for Inflammation Research, 9000 Ghent, Belgium; tineke.vanderhaeghen@irc.vib-ugent.be (T.V.); steven.timmermans@irc.vib-ugent.be (S.T.); 3Department of Biomedical Molecular Biology, Ghent University, 9000 Ghent, Belgium

**Keywords:** sepsis animal models, liver, cecal ligation and puncture, LPS infusion, fecal instillation, RNA sequencing

## Abstract

Sepsis remains a huge unmet medical need for which no approved drugs, besides antibiotics, are on the market. Despite the clinical impact of sepsis, its molecular mechanism remains inadequately understood. Recent insights have shown that profound hepatic transcriptional reprogramming, leading to fatal metabolic abnormalities, might open a new avenue to treat sepsis. Translation of experimental results from rodents to larger animal models of higher relevance for human physiology, such as pigs, is critical and needs exploration. We performed a comparative analysis of the transcriptome profiles in murine and porcine livers using the following sepsis models: cecal ligation and puncture (CLP) in mice and fecal instillation (FI) in pigs, both of which induce polymicrobial septic peritonitis, and lipopolysaccharide (LPS)-induced endotoxemia in pigs, inducing sterile inflammation. Using bulk RNA sequencing, Metascape pathway analysis, and HOMER transcription factor motif analysis, we were able to identify key genes and pathways affected in septic livers. Conserved upregulated pathways in murine CLP and porcine LPS and FI generally comprise typical inflammatory pathways, except for ER stress, which was only found in the murine CLP model. Conserved pathways downregulated in sepsis comprise almost exclusively metabolic pathways such as monocarboxylic acid, steroid, biological oxidation, and small-molecule catabolism. Even though the upregulated inflammatory pathways were equally induced in the two porcine models, the porcine FI model more closely resembles the metabolic dysfunction observed in the CLP liver compared to the porcine LPS model. This comprehensive comparison focusing on the hepatic responses in mouse CLP versus LPS or FI in pigs shows that the two porcine sepsis models generally resemble quite well the mouse CLP model, with a typical inflammatory signature amongst the upregulated genes and metabolic dysfunction amongst the downregulated genes. The hepatic ER stress observed in the murine model could not be replicated in the porcine models. When studying metabolic dysfunction in the liver upon sepsis, the porcine FI model more closely resembles the mouse CLP model compared to the porcine LPS model.

## 1. Introduction

Sepsis is defined as a life-threatening organ dysfunction resulting from a dysregulated host response to infection [1]. Both sepsis and septic shock (i.e., sepsis involving severe blood pressure decline) represent a major global healthcare challenge, being the predominant cause of morbidity and mortality among patients requiring admission to intensive care units (ICUs) and imposing a substantial economic burden on the healthcare system. Recent estimates reveal a staggering annual burden of 48.9 million sepsis cases and 11 million deaths worldwide in the human population [2], but sepsis is also of significant relevance in animals, e.g., in livestock [3]. Based on the epidemiological data, the World Health Organization (WHO) has declared sepsis as a global health priority [4]. The current management of septic patients is rather supportive than curative and relies on controlling the infection using antibiotics, fluid resuscitation combined with vasopressor treatment, and mechanical support of failing organs [5]. Despite five decades of intensive research, no innovative drugs have hit the market to treat sepsis patients [6]. The lack of successful therapies might be attributed to the following issues: (1) For decades, sepsis research has been focused on the inflammatory response, while recent studies have also described the occurrence of metabolic changes in the liver during sepsis [7,8]. (2) Because of practical and ethical constraints, research in sepsis patients is usually limited to serum, plasma, and white blood cells (WBCs). Therefore, it might be of great interest to study organ-specific processes in human sepsis patients; however, it is ethically difficult to obtain biopsies of live patients. In addition, postmortem human biopsies should be interpreted with caution since RNA and protein expression might be altered postmortem [9]. (3) Although rodent models have often yielded valuable information related to certain human diseases, in sepsis, however, translation of preclinical findings to human patients turns out to be difficult, as many promising therapeutic targets in preclinical studies have been disappointing in human clinical studies [6].

Therefore, it is important to close the gap between mouse data and sepsis patients by performing studies with larger animals. In preclinical models, rodents are often used due to their availability, size, low cost, and easy genetic manipulation. Using small animal models, especially transgenic models, allows researchers to gain insight into the proof-of-concept of the molecular mechanisms of diseases such as sepsis in a fast and cost-effective way. In rodent models, sepsis is often induced using cecal ligation and puncture (CLP), which is considered as the gold standard for experimental polymicrobial (peritoneal) sepsis [10]. CLP involves a combination of three insults: (1) tissue trauma due to laparotomy, (2) necrosis caused by ligation of the cecum, and (3) infection due to the leakage of enteric and cecal bacteria into the peritoneum. The latter results in peritonitis followed by translocation of microbial flora into the bloodstream and colonization of organs. It is characteristic of CLP that the pathogens originate from within the host, which is relevant in human septic peritonitis since this usually results from physical intestinal rupture or damage. However, the use of rodents as animal models is also associated with some weaknesses. Due to their small size, it is more difficult to apply regular clinical tools such as probes, catheters, multiple blood samplings, and source control. Moreover, their small size causes a relatively high surface-to-volume ratio, leading to very significant heat loss compared to larger animals and humans. Also, when compared to the human sepsis phenotype, significant differences exist in terms of their metabolic, inflammatory, and hemodynamic responses [11,12]. Therefore, there is a need for more clinically relevant, larger animal models that allow easy and reliable clinical translation [13,14].

The pig genome is related more closely to the human genome compared to the mouse genome, and pigs resemble humans in numerous anatomical and physiological aspects [15]. Due to the similar size of pigs and humans, a preclinical porcine sepsis model can profit from more extensive tools [16,17].

Three categories of experimental sepsis models are used in porcine models, namely (1) the endotoxemia model characterized by lipopolysaccharide (LPS) infusion [18], (2) intravenous administration of live bacterial strains such as *Pseudomonas aeruginosa*, and (3) inoculation of autologous feces (fecal instillation, FI). The use of pigs is, however, associated with higher costs and laborious handling during the experiments. Also, the availability of transgenic pig models is limited [14]. It is crucial to understand that the use of small and large animal models should be seen not as competitive but as complementary. This approach is essential for facilitating the optimal translation of findings from small animal models to large animal models, thereby enhancing the robustness and reliability of preclinical research outcomes.

In the current study, we compared the hepatic transcriptomic changes of CLP mice with those of the two most used porcine sepsis models, namely the porcine LPS and FI model. The CLP (mouse) model is considered the gold standard for studying sepsis, and we aim to determine which porcine model is most suitable for transitioning from CLP mice to pigs, thereby improving translatability in sepsis research.

## 2. Results

### 2.1. Literature Summary of Currently Used Porcine Sepsis Models

Today, the CLP model is considered the gold standard for sepsis studies and is typically induced in mice or rats. Analysis of the porcine sepsis models used in the last five years conducted using ISI Web of Knowledge database (using the query: “*sepsis model pig, article, 2019–2024*”) revealed that the LPS model is the most widely used porcine sepsis model (35%), followed by autologous FI (19%) and monobacterial infections such as *Escherichia coli*, *Streptococcus suis*, and *Pseudomonas aeruginosa* (Appendix A). These porcine models are preferred by researchers, as these are relatively easy to perform—by infusion or instillation—and thus do not require complex surgery on the large animals. The group “other” in the pie chart of Appendix A includes sepsis models such as anastomotic leak, ischemia reperfusion, and abdominal infusion of liver slurry. In this study, the aim was to investigate how closely the hepatic transcriptomic changes in CLP mice align with those observed in the most commonly used porcine sepsis models, specifically the LPS and FI models. Analysis was performed on available datasets, with focus on hepatic gene regulation given our recent observations pointing towards the essential role of the liver and its transcriptomic changes (particularly in relation to nuclear receptor transcription factors) during sepsis [19,20].

To identify differentially expressed genes (DEGs), we used a *p* ≤ 0.05 cutoff and an LFC ≥ 1 or ≤ −1 to determine respectively up- or downregulated genes. Numbers of DEGs detected are summarized in Appendix A. A total of 2019 and 2665 genes are up- and downregulated, respectively, in the mouse CLP model compared to sham mice, whereas the amount of DEGs is lower in the porcine models (683/756 and 761/1235 up/down in, respectively, LPS and FI). This might be attributed to the fact that the annotation of mouse genes is better compared to porcine genes. Indeed, the number of transcripts identified in mice is 2.5× times higher compared to pigs (149.194 gene transcripts are described in *Mus musculus* versus 60.273 in *Sus scrofa*, source: Ensembl). Next, the sepsis severity is higher in mice compared to pigs, as illustrated below. An overview of all DEGs per model and their associated *p*-values and LFCs is provided in Appendix A.

### 2.2. Gene Expression Profiles in Liver from Murine CLP

To comprehensively explore hepatic changes at a genome-wide scale within the livers of septic mice, bulk RNA-seq analysis was conducted on liver tissue obtained 24 h after surgery from mice subjected to either the CLP or sham procedure (Figure 1A,B). To understand the functional implications of the dysregulated genes, a pathway analysis was performed using Metascape [21]. This analysis revealed a typical inflammatory signature being upregulated upon CLP that is characterized by different pathways involved in cell activation, migration, and cell death. Obviously, cytokines are involved, with pathways such as positive regulation of response to external stimulus and positive regulation of cell migration (Figure 1C).

The downregulated genes reveal a profound influence of CLP on diverse metabolic pathways such as monocarboxylic acid metabolism, small-molecule catabolism, biological oxidations, and lipid catabolism (Figure 1D), all key processes within the organism.

To further determine the upstream regulators of the affected genes, HOMER motif analysis was applied. HOMER motif analysis on the upregulated DEGs unveiled factors such as Signal Transducer and Activator of Transcription 3 + Interleukin-21 (STAT3 + IL21), Activator Protein-1 (AP-1), and JUN-AP1 in the top five (Figure 1E), all factors known for their function in humoral immunity, regulation of cell proliferation, differentiation, and apoptosis [22,23,24]. HOMER motif analysis on the downregulated DEGs revealed Hepatic Nuclear Factor 4-alpha (HNF4α) as the top transcription factor being predicted to lose its transcriptional activity upon CLP surgery, followed by Brachyury (TBOX) and Peroxisome Proliferator-Activated Receptor Alpha (PPARα) (Figure 1E). Both HNF4α and PPARα are nuclear receptors known to be involved in processes such as the regulation of lipid metabolism, fatty acid oxidation, and control of energy homeostasis [20,25].

### 2.3. Comparison Analysis: Predicted Common and Unique Affected Pathways in Liver from Murine CLP versus Porcine FI

The porcine model studied here that clinically best resembles the mouse CLP model is the FI model, where, just as in the CLP model, peritonitis is induced through host-derived bacteria. We compared available RNA-seq datasets to explore the similarities and differences in altered pathways between these two septic peritonitis models (Figure 2A). Plotting the log2 fold change (LFC) of all DEGs in CLP mice (*p* ≤ 0.05, LFC ≤ −1 or LFC ≥ 1) versus the LFC of the orthologues genes in the porcine FI model shows a reduced LFC in the porcine FI model. On average, gene expression changes by FI in pigs are limited to 15.17% of the level in CLP mice, as indicated by the slope of the red dotted line (Figure 2B). When overlapping the upregulated DEGs of both murine CLP and porcine FI, an overlap of 27% was found (206/761). Pathway analysis of the shared upregulated genes via Metascape revealed a typical inflammatory response with pathways involved in positive regulation of locomotion, burn wound healing, cytokine signaling in immune system, response to lipopolysaccharide, neutrophil degranulation, and NABA matrisome-associated pathway, the latter being linked to extra cellular matrix (ECM)-associated proteins (Figure 2C).

When overlapping the downregulated DEGs in murine CLP and porcine FI, an overlap of 33% was found (407/1235). Pathway analysis of the shared downregulated genes exclusively displayed an effect on metabolism with pathways involved in small-molecule catabolism, monocarboxylic acid metabolism, biological oxidations, steroid metabolism, and drug metabolism via cytochrome P450 (CYP450) (Figure 2D).

In concordance with the pathway analysis, motif analysis using HOMER of the overlapping upregulated DEGs displayed typical inflammatory-associated transcription factors such as AP-1, FBJ Osteosarcoma Oncogene (FOS), Activating Transcription Factor 3 (ATF3), and JUN, all being basic leucine zipper (bZIP) transcription factors (Figure 2E). HOMER motif analysis of the downregulated genes unveiled three nuclear receptors in the top five, namely HNF4α, PPARα, and Retinoid X Receptor (RXR), all being linked to metabolic pathways (Figure 2E). This comparison analysis shows strong concordance with the results obtained by the CLP only analysis in Figure 1, implicating potential towards a solidified translatability.

In the liver of CLP mice only, 1813 genes are upregulated (Figure 2C). These genes were found to be associated with response to endoplasmic reticulum (ER) stress, apoptotic signaling pathway regulation, regulation of MAPK cascade, and the unfolded protein response (UPR) and are thus more specifically affected in murine CLP. Furthermore, 2258 genes are specifically downregulated in the liver of CLP mice, and the unique pathways identified are, again, predominantly related to metabolism and to vascular processes in the circulatory system (Figure 2D).

Studying the pathways linked to the upregulated genes, specifically in the porcine FI model alone (Figure 2C), again show an inflammatory signature, with pathways involved in response to bacterium, necroptosis, and response to type II interferon. Unique pathways linked to the downregulated genes in the porcine FI model were related to regulation of metabolism of lipids and lipid localization, long-chain fatty acid metabolism, and small-molecule biosynthetic process (Figure 2D). An overview of the overlapping and unique genes shown in the Venn diagrams is provided in Appendix A.

### 2.4. Comparison Analysis: Predicted Common and Unique Affected Pathways in Liver from Murine CLP versus Porcine LPS

In contrast to fecal peritonitis, which is a rather new sepsis induction method, the LPS-induced endotoxemia model is a more routinely used pig model to study sepsis. LPS is derived from the cell wall of Gram-negative bacteria and induces—in contrast to the CLP and FI models—sterile inflammation. While there are studies comparing the transcriptomic response in septic cardiomyopathy between CLP and LPS mice [26], the comparison of the effect of CLP in mice and LPS infusion in pigs on the hepatic gene expression levels has not been investigated so far (Figure 3A). Plotting the log2 fold change (LFC) of all DEGs in CLP mice (*p* ≤ 0.05, LFC ≤ −1 or LFC ≥ 1) versus the LFC of these orthologues genes in the porcine LPS model showed a reduced LFC in the porcine LPS model. On average, gene expression changes by LPS in pigs are limited to 12.14% of the level in CLP mice, as indicated by the slope of the red dotted line (Figure 3B). When overlapping the upregulated DEGs found upon murine CLP and porcine LPS, an overlap of 30% was found (204/683). Pathway analysis of these shared upregulated genes with Metascape revealed a typical inflammatory response very similar to the previous overlap (i.e., CLP-FI) with pathways involved in inflammatory response, NABA matrisome-associated, external stimulus response upregulation, response to bacterium, burn wound healing, and neutrophil degranulation pathways (Figure 3C).

When overlapping the downregulated DEGs in murine CLP and porcine LPS, an overlap of 33% was found (247/756) (Figure 3D). Pathway analysis of the shared downregulated genes again displayed an effect on metabolism with pathways involved in monocarboxylic acid metabolism, steroid metabolism, biological oxidations, small-molecule catabolism, and the nuclear receptors meta-pathway. Motif analysis using HOMER of the overlapping upregulated DEGs displayed typical inflammatory-associated transcription factors such as RUNX, NF-κB-p65-Rel, and TEALD1. Motifs of the overlapping downregulated genes point towards Nuclear Factor (NF)1, HNF4α, and HNF1β (Figure 3E).

Analyzing the DEGs upregulated in the liver of CLP mice only, 1815 genes were identified (Figure 3C). These genes were found to be associated with response to ER stress, cytokine signaling in the immune system, positive regulation of cell migration, and UPR response, quite similar to the unique pathways identified in CLP mice when compared to the FI model. In contrast, 2418 genes are specifically downregulated in the liver of CLP mice, and the unique pathways identified are associated with lipid catabolism, vascular process in circulatory system, nucleobase-containing small-molecule metabolism, organic hydroxy compound metabolism, and the long-chain fatty acid metabolic process (Figure 3D).

Studying pathways linked to the upregulated DEGs specifically in the porcine LPS model, an inflammatory signature with positive regulation of cytokine production and innate immune response as well as metabolic pathways involved in amino acid metabolism were detected (Figure 3C). Unique pathways linked to the downregulated genes, specifically in the porcine LPS model, are related to metabolic pathways such as cholesterol metabolism with Block and Kandutsch Russell pathways, fatty acid metabolism, steroid biosynthesis, and NABA core matrisome (Figure 3D). An overview of the overlapping and unique genes shown in the Venn diagrams is provided in Appendix A.

### 2.5. Comparison Analysis: Predicted Common and Unique Affected Pathways in Liver from Porcine FI versus Porcine LPS

To determine the common and unique pathways being affected in the two porcine animal sepsis models, we compared the DEGs upon porcine FI and porcine LPS (Appendix A). To date, the hepatic gene response upon FI versus LPS in pigs has not yet been compared. An overlap of 35% was found (239/683) in the upregulated genes and 48% (360/756) in the downregulated genes (Appendix A). These percentages are higher compared to the previous two comparison analysis with the murine CLP model and might be attributed to the higher amount of gene transcripts defined in mice compared to pigs. In the shared upregulated genes, the most important pathways identified are related to inflammation, characterized by cellular response to cytokine stimulus, inflammatory response, the NABA matrisome-associated pathway, neutrophil degranulation, and cytokine signaling in the immune system (Appendix A). Pathway analysis of the shared downregulated genes again revealed a major impact on steroid, monocarboxylic acid metabolism, and biological oxidations, just as was observed in the pairwise analysis with the murine CLP model (Appendix A). HOMER motif revealed, e.g., FOS and JUNB in the analysis of the common upregulated genes between porcine FI and LPS (Appendix A). More significant differences were detected in the transcription factor hits of the shared downregulated genes. With the exception of HNF1β, V-maf Musculoaponeurotic Fibrosarcoma oncogene homolog B (MafB) and Broad complex-tramtrack-bric a brac And Cap‘n’collar Homology 2 (BACH2) are newly identified transcription factors (Appendix A), known to play a role in immune-related cells such as macrophages, monocytes, T cells, and B cells [27,28]. It is very striking that practically all motifs found in the porcine comparison analysis are bZIP transcription factors, which are one of the largest families of transcription factors. They are widely distributed and highly conserved in animals, plants, and microorganisms. All bZIP transcription factors contain leucine zippers that enable homo- or heterodimerization.

The FI model has the most unique DEGs among the two porcine models, especially when studying the downregulated genes. In total, 522 genes were found to be upregulated in the FI model only, and these genes belong to positive regulation of cell motility, actin filament-based process, nuclear receptors meta-pathway, and import into cell and tissue morphogenesis (Appendix A). In contrast, 875 genes are specifically downregulated in the liver of FI pigs, and the unique pathways identified are associated with organic acid biosynthetic process, glycine, serine and threonine metabolism, amino acid metabolism, and the PPAR signaling pathway (Appendix A). The unique pathways found to be upregulated in the porcine LPS model have a strong inflammation signature, whereas the 396 unique genes downregulated in the LPS porcine model are found to be involved in small-molecule biosynthetic process, SREBP signaling, fatty acid metabolic process, and lipid homeostasis (Appendix A). An overview of the overlapping and unique genes shown in the Venn diagrams is provided in Appendix A.

### 2.6. Side-by-Side Comparison of Affected Pathways Based on Target Gene Expression in the Three Different Sepsis Models

In previous comparison analysis, the three sepsis models were evaluated at the pathway level, focusing solely on significantly altered genes. This approach offers a general understanding of the types of pathways affected but does not indicate the extent of their alteration. In order to know to what extent the pathways in murine CLP are changed (shown in Figure 1C; comparable to the Z-score in Ingenuity Pathway Analysis (IPA)), the genes behind these pathways according to Metascape were retrieved.

By plotting the LFC of the affected genes within each pathway, we can better understand the extent to which each pathway is impacted across the models. For this analysis, only genes with a gene annotation present in all three models were included. Figure 4A reveals that for the top five upregulated pathways retrieved from the murine CLP model (see also Figure 1C), the CLP model induces the different gene sets the strongest, with an average LFC of ± 3 per pathway. Both the LPS and FI porcine models also induce these pathways but with an average LFC of ± 0.6 per pathway. Since these upregulated pathways are typically associated with inflammation, the data suggest that the CLP model induces the strongest inflammatory response, while the two porcine models exhibit a similar level of inflammation to each other, though it is less intense than that observed in the murine model.

When comparing the LFC of gene sets retrieved from the downregulated pathways detected in the CLP model (see also Figure 1D), the murine CLP model again has the strongest effect with an average LFC of ± −2.5 (Figure 4B). Interestingly, for each pathway that was found to be downregulated in the murine model, the FI model (average LFC ± −1) has a stronger effect compared to the LPS model (average LFC ± −0.5). As these downregulated pathways are typically related to metabolic functions, we can conclude from these data that the FI model is a better model to mimic the CLP-induced metabolic dysfunction in the liver compared to the porcine LPS model.

An illustration of these findings is the LFC of typical genes involved in metabolism, which are dependent on HNF4α or PPARα—two transcription factors predicted (Figure 1F) and shown to lose function in the murine liver following CLP surgery [20,29]. HNF4α is a transcriptional regulator of hepatocyte identity and controls genes that are essential for liver functions, such as *Cyp7a*, *Hes6*, *Nr1h4*, *Ppara*, and *Apoa2* (Figure 4C). All these genes were found to be downregulated in murine liver upon CLP compared to sham, and a similar trend was observed in the porcine FI model. The downregulation of these target genes is, however, less apparent in the porcine LPS model. Next, we focused on the target genes of PPARα (Figure 4D). PPARα is known to be involved in β-oxidation of free fatty acids (FFA) and ketogenesis by inducing, e.g., *Ehhadh*, *Hmgcs2*, *Hadha*, *Slc25a20*, *Cpt2*, and *Cpt1a*. Again, all these genes were found to be downregulated in murine liver upon CLP compared to sham, and a similar trend was observed in the porcine FI model, but no clear downregulation was observed in the porcine LPS model. *Hmgcs2*, known to be involved in ketogenesis, is even significantly upregulated upon LPS in the porcine model, a finding also observed in the murine LPS model. These data are in concordance with the HOMER analysis on the downregulated genes shown in Figure 2E and Figure 3E, showing a stronger enrichment of these nuclear receptors in the FI model compared to the LPS model.

Remarkably, the pathways “response to ER stress” and “UPR response” are uniquely affected among the upregulated genes in the mouse CLP model. These pathways were not identified by Metascape when analyzing the DEGs in either of the porcine models. When plotting the LFC of typical ER stress- and UPR-related genes (Figure 4E), we observe a strong upregulation of these genes in the murine sepsis model, a less pronounced induction in the porcine LPS model, and a decrease in gene expression in the FI porcine model. In contrast, one of the pathways uniquely affected amongst the downregulated genes in the mouse CLP model is “vascular process in circulatory system”. This pathway was not retrieved by Metascape when entering the DEGs of either porcine model. When plotting the LFC of some typical genes involved in vascular function, as provided by Metascape, e.g., genes involved in regulation of endothelial permeability (*Angpt1*, *Tjp3*, *Tek*, and *Ocln*), platelet aggregation (*Tbxa2r*), transport of metabolites (*Slc* genes), complement activation (*Crp*), vascular tone (*Arhgap42*), antioxidant activity (*Sod3* and *Gpx1*), and nitric oxide generation (*Ddah1*), a strong downregulation of the genes in murine sepsis was observed. The two porcine models show similar behavior to each other in terms of LFC; however, unlike the CLP model, there is no clear downregulation of the described genes (Figure 4F). As vascular function is heavily affected by inflammation, the more pronounced effect of the murine CLP model on these vascular-related genes could be attributed to the higher inflammation that is detected in the mice. Of note, due to ethical considerations, there is a slight variation in timepoints amongst the different groups. To verify the robustness of the data, we also compared the LFC of the described genes in Figure 4C–F with another publicly available CLP dataset. This dataset was first described in Vandewalle et al. [19] and compared the hepatic transcriptome in sham or CLP mice 8 h after surgery. Interestingly, we confirmed that the genes depicted in the 24 h CLP dataset that were, respectively, up- (ER stress, Figure 4E) or downregulated (HNF4α, PPARα, and vascular process-related genes, Figure 4C,D,F) show a similar trend in the 8 h CLP dataset (Appendix A). These data point towards the robustness of the data, irrespective of the timepoint chosen.

Overall, side-by-side comparisons between the three different animal models give a comparable overlap of genes shared between the murine CLP versus the two porcine sepsis models. The upregulated genes are typically associated with inflammatory processes, whereas the downregulated genes strongly indicate metabolic dysfunction in the liver. Upregulated genes that are not shared in the comparative analysis are similarly associated with inflammation and vice versa; downregulated genes that are not shared in the comparative analysis are still indicative for metabolic dysfunction. The murine CLP model has the strongest effect on both the up- and downregulated pathways. The two porcine models have a milder effect on the described pathways, with the porcine FI model most closely resembling the downregulated metabolic pathways found in the CLP model.

## 3. Discussion

Despite extensive research on sepsis, there is no real advancement in sepsis treatment, and many drugs have failed to show a clinical effect in human trials despite being successful in preclinical animal models [6]. Even worse, some therapies, such as TNFα neutralizing antibodies, were promising in preclinical studies but appeared to be harmful in sepsis patients. The animal model used to study drug efficacy has proven to be of utmost importance. Indeed, anti-TNF therapy reduced mortality in the LPS model, whereas TNF turned out to be essential for controlling the infection in the CLP model [30]. A similar disparity was observed when using TNFR1KO mice in murine LPS and CLP models [31]. Due to high similarities between the CLP model and human sepsis, the CLP model is regarded as the gold standard for studying sepsis [10]. Another major concern regarding the current sepsis research is that most studies examine the inflammatory aspect, with focus on serum, plasma, and WBCs. Of course, obtaining blood samples or determining blood parameters is the easiest way to study sepsis in human patients, but rather than investigating what happens in the blood, we need to focus on what happens in the organs, with the liver as the most promising organ based on our most recent insights [8,19,20]. The liver hosts, next to hepatocytes, a range of other cells, including endothelial, hepatic stellate, and Kupffer cells, which all play an essential role in a wide range of cellular processes, such as homeostasis, metabolism, and immunity. Some research has been conducted on postmortem liver samples of sepsis patients, but analysis of postmortem material is difficult owing to changes that occur within the body shortly after death [9]. Therefore, studies on animals such as mice and rats are indispensable, but to enhance translatability, research is warranted in larger animals that are more closely related to humans. Compared to the mouse genome, the pig genome is three times closer to that of a human, and its anatomy and physiology are also more comparable to humans [15]. Indeed, the macroscopic structure (subdivision in lobes and segments) and vascularization are comparable between pig and human livers. Unlike rodents, pigs have well-defined acinar structures with clear portal triads, similarly to humans [32]. In the current study, we examined whether the liver-specific gene expression profile in a murine CLP model reflects the profile seen in the two most routinely used porcine sepsis models. We sought to determine which porcine model is most suitable for follow-up studies based on findings from mouse research. Therefore, the hepatic transcriptomic changes of CLP mice were compared to the LPS and the FI porcine sepsis models. Analysis was carried out using Metascape, as this tool is very user friendly to analyze OMICs data.

A first view on the amount of DEGs (set as *p* ≤ 0.05 and a LFC ≥ 1 or ≤−1) showed 4684 DEGs in CLP mice compared to sham mice, whereas for the porcine LPS and FI model, this was, respectively, 1439 and 1996 genes. This reflects the higher number of transcripts identified in mice compared to pigs but could also be explained by the reduced variation and stronger effect size in mice. Throughout our comparative analysis, the major upregulated pathways point towards inflammation. However, the type of pathways activated in the different models used might differ. Unique to the CLP model is a strong response to ER stress and UPR, whereas this could not be detected in the liver of the two porcine models. The ER is involved in protein folding and post-translational modifications and is also a critical organelle of the secretory pathway. The burst of protein synthesis during the early phase of sepsis can lead to the accumulation of unfolded or misfolded proteins, a phenomenon also known as ER stress. ER stress can eventually activate the UPR response, an adaptive reaction that reduces unfolded protein load to maintain cell viability and function [33]. A closer look at typical ER stress-related genes indeed show a strong upregulation of these genes in the murine CLP model, whereas in the porcine LPS model and especially the porcine FI model, this was less apparent. This may indicate that the porcine model is not a good model to study hepatic ER stress during sepsis when making the transition from mice to larger sepsis models. In general, there are not many studies investigating the role of ER stress in liver during sepsis, and if so, these studies were performed in rodents. ER stress is activated upon sepsis in rodents, and inhibition of ER stress using 4-phenylbutyric acid could improve vital organ function in the liver, kidney, and intestinal barrier, resulting in improved survival following CLP in rats [34]. ER stress in septic pigs has not yet been reported in the literature. A potential reason for the lack of depiction of a strong ER stress response in porcine sepsis could be the comparatively lower inflammatory response observed in pigs compared to the CLP mouse model, as the inflammatory response and ER stress are strongly linked to each other. Indeed, it is known that inflammatory cytokines such as the pro-inflammatory cytokines IL1β, IL6, and TNFα induce ER stress, which further disrupts metabolic functions, thereby causing more inflammation. This vicious cycle exacerbates inflammatory signaling as well as metabolic deterioration [35]. Of note, compared to mice, the immune system of pigs generally more strongly resembles that of humans. Indeed, the lymphocyte % in blood is 75–90% in mice, 40–60% in pigs, and 20–50% in humans, whereas the neutrophil % in blood is 10–25% in mice, 30–50% in pigs, and 40–75% in humans [17,36]. These differences in immune composition might account for the stronger immune response to sepsis compared to pigs. In human sepsis patients, ER stress in skeletal muscle and immune cells has already been depicted [37]; yet, whether ER stress plays a role in the liver of human sepsis patients needs to be established. Based on the current data, it remains uncertain whether inhibiting ER stress would yield significant therapeutic benefits for human sepsis patients. While preclinical studies in mice suggest a role for ER stress in the pathophysiology of sepsis, translating these findings to humans is complex due to interspecies differences in immune response and disease progression. Further studies, including clinical trials, are needed to evaluate the potential efficacy and safety of ER stress inhibition in sepsis treatment.

Interestingly, the major downregulated pathways throughout our comparison analysis almost exclusively involve metabolic pathways. This observation supports recent findings showing that upon CLP surgery in mice, the liver loses activity of key transcription factors involved in metabolism [8], such as glucocorticoid receptor (GR) [19], PPARα [20], and HNF4α [29], eventually leading to metabolic failure. Indeed, upon CLP surgery, the liver acquires GR resistance. Since GR is involved in gluconeogenesis, GR functioning is essential for sepsis survival due to its role in lactate removal and glucose production [19]. The “monocarboxylic acid metabolism” pathway was found amongst the downregulated genes in all three models tested. Typical monocarboxylic acids includes lactate, pyruvate, and ketone bodies, and these metabolites are indeed all found to be increased upon sepsis. The GR motif was, however, not found by HOMER amidst the tested downregulated genes, which might be attributed to the initial corticosterone boost shortly after sepsis that first induces many GR-responsive genes [19]. PPARα is another major transcription factor involved in metabolism by coordinating the β-oxidation of FFAs in liver. Similar to that observed with GR, PPARα loses its transcriptional activity upon CLP surgery in mice, causing FFAs to accumulate in blood and organs eventually leading to lipotoxicity [20]. This finding was already confirmed in the porcine FI model [38], and in the current study, the porcine LPS model also presented with “fatty acid metabolic process” amongst the pathways retrieved from the downregulated genes. The PPARα motif was found amidst the downregulated DEGs of the murine CLP and porcine FI model according to HOMER but not in the porcine LPS model. In accordance with HOMER, *Ppara* gene expression and its target genes were less clearly downregulated upon LPS injection in pigs compared to the murine CLP and porcine FI model. This might explain why the “lipid catabolic process” is the least affected in the porcine LPS model compared to the two other models tested. Recently, it was demonstrated that PPARα mRNA and protein levels are downregulated upon sepsis due to HNF4α loss of function [29]. HNF4α is key regulator of hepatocyte identity by controlling genes involved in metabolism of lipids, glucose, bile acids, and xenobiotics. Transgenic mice with HNF4α-specific deletion in hepatocytes have severely reduced hepatic PPARα levels in steady state. Upon sepsis in wild-type mice, HNF4α is shown to lose its function, leading to PPARα reduction [29]. Next to its role in controlling PPARα expression and lipid metabolism, HNF4α also induces many CYP450-coding genes. CYP450 enzymes, encoded by the *Cyp* genes, are involved in the synthesis and metabolism of various compounds within cells. CYP450 enzymes are key phase I enzymes that metabolize many xenobiotics, such as drugs and environmental pollutants, and endogenous substances, such as toxins that are formed within cells. Interestingly, “drug or xenobiotic metabolism by cytochrome P450” was found by Metascape amongst the downregulated genes in all three sepsis models tested, and the HNF4α motif was also found amidst the downregulated genes in all three models. That metabolism reduced by CYP450 is a consequence of HNF4α loss of function seems logical but needs to be confirmed. Given the high similarities between pigs and humans in substrate selectivity and quantity-normalized activities for major drug-metabolizing CYP450 subfamilies in the liver [39], the porcine FI model thus seems a very good model when making the transition from mice to humans in studying CYP450 activity in septic liver. Just as observed with PPARα-dependent genes, other HNF4α target genes were also generally downregulated in both the murine CLP and porcine FI models, whereas the downregulation of these genes was less apparent in the porcine LPS model. This observation could be expanded to the top five downregulated pathways found in CLP mice, implicating that the FI model is a better model to mimic metabolic dysfunction in CLP mice compared to the porcine LPS model. Next to the global metabolic signature observed amongst the downregulated genes, “vascular process in circulatory system” was a typical pathway observed in the murine CLP model and not in the two porcine models when the downregulated DEGs were entered in Metascape. This might be related to the higher inflammation signature observed in the CLP mice compared to the two other models, as inflammation is a well-known inducer of vascular dysfunction.

The current study has some limitations. First, we made use of two published datasets to perform the comparative analysis in order to avoid unnecessary repetition of experiments, with only minor differences in isolation timing, in accordance with ethical considerations. This implies that the timepoints of liver collection are not exactly the same. For the murine CLP model, this was 24 h, whereas for the porcine LPS and FI model, this was, respectively, 15 h and 18 h after sepsis induction. Despite the later timepoint in the murine model, we had a greater inflammatory response in the murine model compared to the porcine models. This was inherent to the mouse model and not to the timepoint, as we observed the same finding when comparing the porcine datasets with our available 8 h CLP dataset. Interestingly, despite the slightly different timepoints between the two porcine models, the (inflammatory) upregulated pathways were equally induced in both models, suggesting a same degree of inflammation and sepsis severity. Second, it would have been interesting to compare sepsis severity not only based on transcriptional induction of inflammatory pathways but also based on the Sequential Organ Failure Assessment (SOFA) score. Clinical markers to score sepsis severity were, however, not measured in the porcine LPS model and murine CLP model, making it impossible to correlate the degree of affected pathways with sepsis severity. Third, the critical care provided to the animals was different amongst the three models. For the murine CLP model, this entailed antibiotics and fluid administration; for the porcine LPS model, no critical care was provided; and for the porcine FI model, this entailed an advanced intensive care that follows the MQTiPSS (Minimum Quality Threshold in Pre-Clinical Sepsis Studies) recommendations (including abdominal lavage, antimicrobial therapy, and early vasopressor introduction) in order to mimic human clinical care [40]. Lastly, we used the murine CLP model as the gold standard to study sepsis, but of course, ideally, we would have compared transcriptomic changes in the liver of animal models with those observed in human sepsis livers. We are currently conducting a clinical study to collect liver biopsies from (live) human peritoneal sepsis patients as well as controls to explore whether the observed results also apply to human patients (approved by ethical committee UZ Ghent (ONZ-2022-0520)).

Collectively, the pairwise analysis between the murine CLP–porcine LPS, murine CLP–porcine FI and porcine LPS–porcine FI generally showed an inflammatory pathway amongst the upregulated genes and metabolic failure amongst the downregulated genes. Focusing on the inflammatory pathways, ER stress and the UPR response were found to be induced in the liver of CLP mice, but this could not be confirmed in either of the porcine models. This might imply that ER stress does not occur in the porcine liver and—by extension—perhaps human sepsis livers. Focusing on the downregulated metabolic pathways, it was notable that the porcine FI model more closely resembles the murine CLP model than the porcine LPS model. Even though the LPS model is the most widely used sepsis model in pigs, it is thus advisable to use the porcine FI model, especially when studying the metabolic dysfunctions occurring in the liver following sepsis.

## 4. Materials and Methods

### 4.1. Animals

#### 4.1.1. Mice

Male C57BL/6J mice at 8–12 weeks of age were purchased from Janvier (Le Genest-St. Isle, France). Mice were housed in a temperature-controlled, specific-pathogen-free (SPF), air-conditioned animal house with 14 h and 10 h light/dark cycles and received food and water ad libitum. All experiments were approved by the institutional ethics committee for animal welfare of the Faculty of Sciences, Ghent University, Belgium.

#### 4.1.2. Pigs—LPS Infusion

Eight-week-old male pigs (Landrace × Large White, Seghers Hybrid^®^, RA-SE Genetics, Lokeren, Belgium) were group-housed in standard stables (2.30 m × 2.40 m) and provided with water and feed ad libitum. Throughout the entire experiment, the stable temperature was maintained at an average of 23.2 ± 0.45 °C, and the stables were enriched with rubber toys and cotton towels. The LPS infusion study was approved by the Ethical Committee of the Faculty of Veterinary Medicine and the Faculty of Bioscience Engineering of Ghent University (EC 2017/24).

#### 4.1.3. Pigs—Fecal Instillation

Twelve pigs (seven male and five female, weighing 49 ± 5 kg) (RA-SE Genetics, Belgium), were kept in standard stables and provided with water and feed ad libitum. The study protocol followed the EU Directive (2010/63/EU) for animal experiments and was approved by the local animal ethics committee (Comité Ethique du Bien-Être Animal; protocol number 724N) from the Université Libre de Bruxelles (ULB) in Brussels, Belgium. These pig experiments were conducted in the Experimental Laboratory of Intensive Care of ULB (LA1230406), and the Minimum Quality Threshold in Preclinical Sepsis Studies (MQTiPSS) guidelines recommendations and ARRIVE guidelines for translational sepsis research were followed [40,41].

### 4.2. Experimental Procedures

#### 4.2.1. Cecal Ligation and Puncture (CLP)

A detailed description of the experimental procedure to induce polymicrobial sepsis in mice was provided by Vandewalle et al. (2021) [19]. In short, polymicrobial sepsis was induced in mice using the CLP technique. For this, mice were anesthetized with isoflurane, and a midline abdominal incision was made to expose the cecum. Then, the cecum was ligated for 75%, and a double puncture using a 21-gauge needle was made. The abdominal muscles were sutured with running stitches, and the skin was closed using metallic clips. As a control group, the cecum was exposed in sham mice but neither ligated nor punctured. Mice were euthanized via cervical dislocation 24 h or 8 h (as detailed in text) after the onset of sepsis, and liver samples were collected for subsequent analysis.

#### 4.2.2. LPS Infusion

A detailed description of the experimental procedure to induce endotoxemia in the pigs was provided by Dhondt et al. (2021) [42]. In short, following a 3-day acclimatization period, a double-lumen catheter was surgically implanted in the external jugular vein of piglets. After surgery, piglets were housed individually to prevent catheter displacement. To maintain catheter patency, regular flushing with a heparinized solution was performed, and bandages were replaced daily. Endotoxemia was induced after a recovery day, during which piglets received a continuous infusion of ultrapure LPS derived from *Escherichia coli* (O111) (Invitrogen, Toulouse, France) over a 15 h period at a rate of 5.0 µg/kg/h, alongside fluid therapy consisting of 0.9% NaCl at 6 mL/kg/h. Sham piglets were administered only the vehicle solution. After 15 h of LPS infusion, euthanasia was induced by an overdose of sodium pentobarbital (Sodium pentobarbital 20%^®^, Kela, Hoogstraten, Belgium), after which liver samples were collected for subsequent analysis.

#### 4.2.3. Fecal Instillation

A detailed description of the experimental procedure to induce septic shock in the pigs was provided by Garcia et al. (2022) [43]. In short, prior to the start of the experiment, pigs were fasted for 18 h with free access to water. Fecal peritonitis resulting in septic shock was induced in 9 pigs (5 males and 4 females) by intraperitoneal instillation of 3 g/kg of autologous feces collected from the pig’s enclosure and diluted in 300 mL of 5% glucose solution. Three sham pigs (two males and one female), consisting of anesthesia and surgical preparation without sepsis induction, were used as a control. The onset of septic shock was set at a mean arterial pressure (MAP) below 50 mmHg. A MAP between 45 and 50 mmHg was then maintained for 60 min to consolidate the multiple organ failure induction. Resuscitation fluids, norepinephrine (NE), antibiotics treatment, and abdominal lavage were applied to mimic human clinical care during the following 8 h. After 18 h of FI infusion, pigs were euthanized with 7.5% potassium chloride under deep anesthesia for organ isolation. Liver samples were isolated from the NE group at the end of the experiment (vasopressor 2 timepoint) for subsequent analysis.

#### 4.2.4. RNA Sequencing

##### Mouse Liver—CLP Dataset

We used the liver murine CLP datasets GSE160795 (8 h timepoint) and GSE160830 (24 h timepoint) that was processed as described in Vandewalle et al. (2021) [19].

##### Pig Liver—LPS Dataset

Liver biopsies from the fourth segment were preserved in RNALater (AM7021, Invitrogen, Waltham, MA, USA) and subsequently processed for RNA extraction using the Aurum total RNA mini kit (732-6820, Bio-Rad, Temse, Belgium), following the manufacturer’s protocol. The concentration and quality of the extracted RNA were analyzed using Agilent RNA 6000 Pico Kit (Agilent Technologies, Santa Clara, CA, USA). The RNA samples were sequenced by VIB Nucleomics Core (Leuven, Belgium). Libraries were prepared using the Illumina TruSeq3 stranded library prep protocol, and sequencing was performed with single-end reads (75 bp) on an Illumina NextSeq 550 platform. Pig reads were aligned to the Sscrofa11 reference genome using STAR 2.7.10a with the genequant option for direct read-count assignment to features. Differential gene expression analysis was conducted with DESeq2, and results were exported in Excel 2019 format. The RNA-seq data were deposited in the Gene Expression Omnibus (GEO) under accession number GSE250039 and are publicly accessible via the NCBI database.

##### Pig Liver—Fecal Instillation Dataset

We used the liver porcine FI dataset GSE218636 that was processed as described in Vandewalle et al. (2022) [38].

#### 4.2.5. Processing of the RNA-Seq Results

The dataset under investigation was acquired in the form of Excel spreadsheets containing the genes associated with each sepsis model. To ensure the acquirement of statistically significant outcomes, we applied filters: adjusted *p*-value ≤ 0.05, log fold change (LFC) ≥ 1 for upregulated genes, and LFC ≤ −1 for downregulated genes. A comparative analysis between distinct sepsis models (CLP mouse versus porcine LPS, CLP mouse versus porcine FI, and porcine FI versus porcine LPS) was conducted. A considerable proportion of the genes associated with porcine sepsis lacked a designated name corresponding to their ENSG ID. To address this, genes lacking a specific name were filtered out. The refined gene lists were subsequently subjected to Ensembl Biomart analysis. We specifically focused on the Ensembl 110 genes database schema, selecting the dataset related to pig genes (Sscrofa11.1). Our criteria for this analysis included filtering by gene-stable ID (e.g., ENSSSCG00000000002) and extracting selected attributes such as homologues (up to six orthologues). The focus was on mouse orthologues, where we set our filters to receive the mouse gene names respective to the ENSG ID. The final gene lists were compared using Venn diagrams, juxtaposing the gene lists for distinct sepsis models (CLP mouse versus porcine LPS, CLP mouse versus porcine FI, and porcine FI versus porcine LPS). Each list was labeled with its corresponding designation and processed for intersection and independent region outcomes. These results were further subjected to pathway analysis using the Metascape analysis interface [21] (v3.5.20240101) with input as species “any species” and analysis as species “*H. sapiens*” and otherwise default settings. Motif finding for multiple motifs was carried out using the HOMER software version v4.11 [44]. We used the promoter region (start offset: −1 kb, end offset: 50 bp downstream of the transcription start site (TSS)) to explore known motif enrichment. HOMER motif enrichment *p*-values were determined using the whole genome promoter set as background. The reported *p*-values are those of the motif overrepresentation of the gene set of interest over background for each motif.

## Figures and Tables

**Figure 1 ijms-25-11079-f001:**
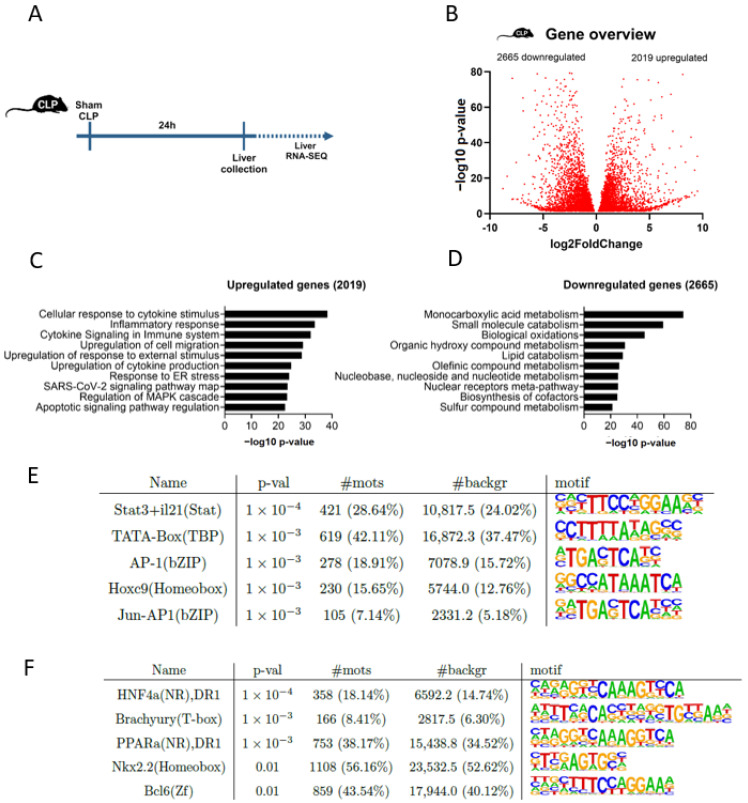
Genome-wide overview of the gene expression profile of the liver in CLP-induced polymicrobial sepsis in mice. (**A**) Male C57BL/6J mice were subjected to a sham or CLP procedure, and the livers were isolated 24 h after surgery for a genome-wide transcriptomics analysis via bulk RNA-seq. (**B**) Volcano plot depicting the differentially expressed genes (*p* ≤ 0.05, LFC ≤ −1 or LFC ≥ 1) affected by CLP compared to sham. Genes with a −log10 *p*-value exceeding 80 were excluded for improved plot clarity. (**C**,**D**) Top ten enriched gene ontology (GO) terms for genes that are upregulated (*p* ≤ 0.05, LFC ≥ 1) (**C**) or downregulated (*p* ≤ 0.05, LFC ≤ −1) (**D**) in CLP mice compared to sham controls. Analyses were performed using the Metascape analysis interface. (**E**,**F**) HOMER motif analysis of CLP-induced genes (**E**) or genes downregulated by CLP (**F**) compared to sham controls (start offset: −1 kb, end offset: 50 bp downstream TSS) (*p* ≤ 0.05). Enriched motifs with their name and *p*-value (*p*-val) are displayed. #mots = number of motifs found amongst the downregulated genes (absolute (relative %)); #background = number of motifs found amongst background genes (absolute (relative %)).

**Figure 2 ijms-25-11079-f002:**
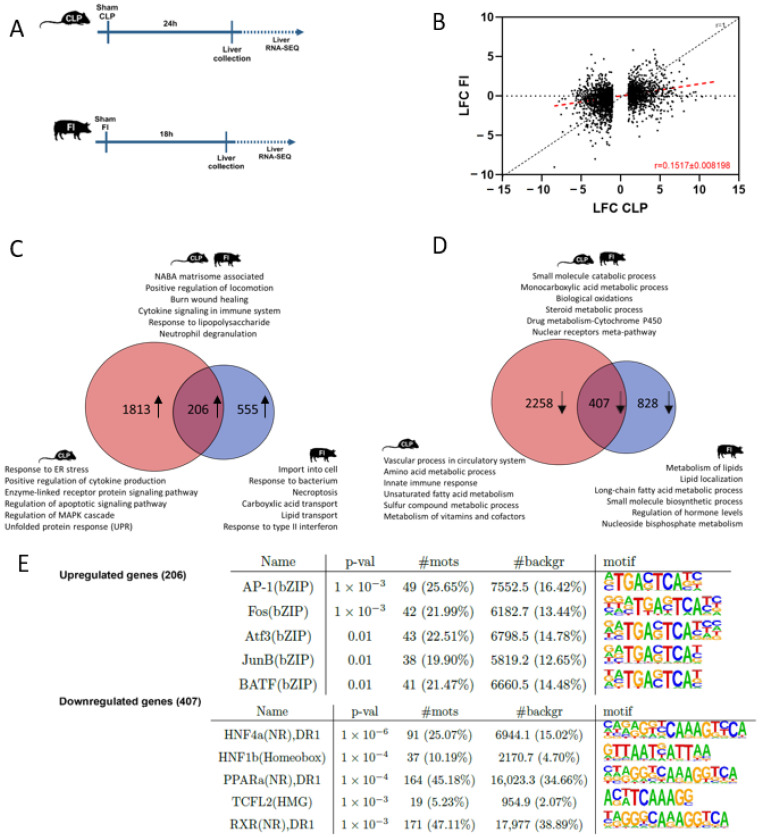
Comparative analysis of the gene expression profile in the liver of CLP-induced polymicrobial sepsis in mice and FI-induced polymicrobial sepsis in pigs. (**A**) Experimental set-up. Mice were subjected to a sham or CLP procedure, and pigs were subjected to a fecal peritonitis (FI) or control procedure. Livers were isolated, respectively, 24 h and 18 h after initiation for genome-wide transcriptomics analysis using bulk RNA-seq. (**B**) Scatter plot demonstrating LFC of all differentially expressed genes in CLP mice (*p* ≤ 0.05, LFC ≤ −1 or LFC ≥ 1) versus the LFC of the orthologues genes in FI pigs. r = 1 depicts the slope if no difference in LFC can be observed between CLP or FI. The red dotted line indicates the real slope of the data. Data were analyzed with linear regression. (**C**) Venn diagram depicting the overlap between the upregulated DEGs in murine CLP versus porcine FI. Selected condition-specific enriched Metascape pathways and biological functions are noted. For the separate DEGs in CLP or FI, only the unique pathways and biological functions are depicted. (**D**) Venn diagram depicting the overlap between the downregulated DEGs in murine CLP versus porcine FI. Selected condition-specific enriched Metascape pathways and biological functions are noted. For the separate DEGs in CLP or FI, only the unique pathways and biological functions are depicted. (**E**) HOMER motif analysis of the shared up- and downregulated DEGs in CLP mice and FI pigs compared to their appropriate controls (start offset: −1 kb, end offset: 50 bp downstream TSS) (*p* ≤ 0.05 and genes with mouse orthologue). Enriched motifs with their name and *p*-value are depicted. #mots = number of motifs found amongst the downregulated genes (absolute (relative %)); #background = number of motifs found amongst background genes (absolute (relative %)).

**Figure 3 ijms-25-11079-f003:**
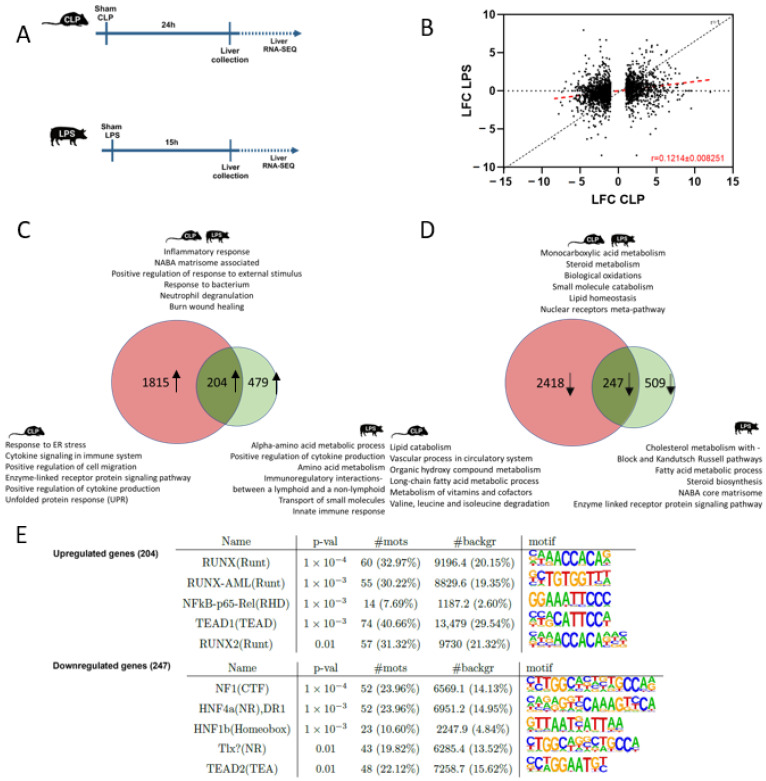
Comparative analysis of the gene expression profile in the liver of CLP-induced polymicrobial sepsis in mice and LPS-induced endotoxemia in pigs. (**A**) Experimental set-up. Mice were subjected to a sham or CLP procedure, and pigs were subjected to a LPS infusion or control procedure. Livers were isolated, respectively, 24 h and 15 h after initiation for genome-wide transcriptomics analysis using bulk RNA-seq. (**B**) Scatter plot demonstrating LFC of all differentially expressed genes in CLP mice (*p* ≤ 0.05, LFC ≤ −1 or LFC ≥ 1) versus the LFC of these orthologues genes in LPS pigs. r = 1 depicts the slope if no difference in LFC can be observed between CLP or LPS. The red dotted line indicates the real slope of the data. Data were analyzed with linear regression. (**C**) Venn diagram depicting the overlap between the upregulated DEGs in murine CLP versus porcine LPS. Selected condition-specific enriched Metascape pathways and biological functions are noted. For the separate DEGs in CLP or LPS, only the unique pathways and biological functions are depicted. (**D**) Venn diagram depicting the overlap between the downregulated DEGs in murine CLP versus porcine LPS. Selected condition-specific enriched Metascape pathways and biological functions are noted. For the separate DEGs in CLP or LPS, only the unique pathways and biological functions are depicted. (**E**) HOMER motif analysis of the shared up- and downregulated DEGs in murine CLP versus porcine LPS compared to their appropriate controls (start offset: −1 kb, end offset: 50 bp downstream TSS) (*p* ≤ 0.05 and genes with mouse orthologue). Enriched motifs with their name and *p*-value are depicted. #mots = number of motifs found amongst the downregulated genes (absolute (relative %)); #background = number of motifs found amongst background genes (absolute (relative %)).

**Figure 4 ijms-25-11079-f004:**
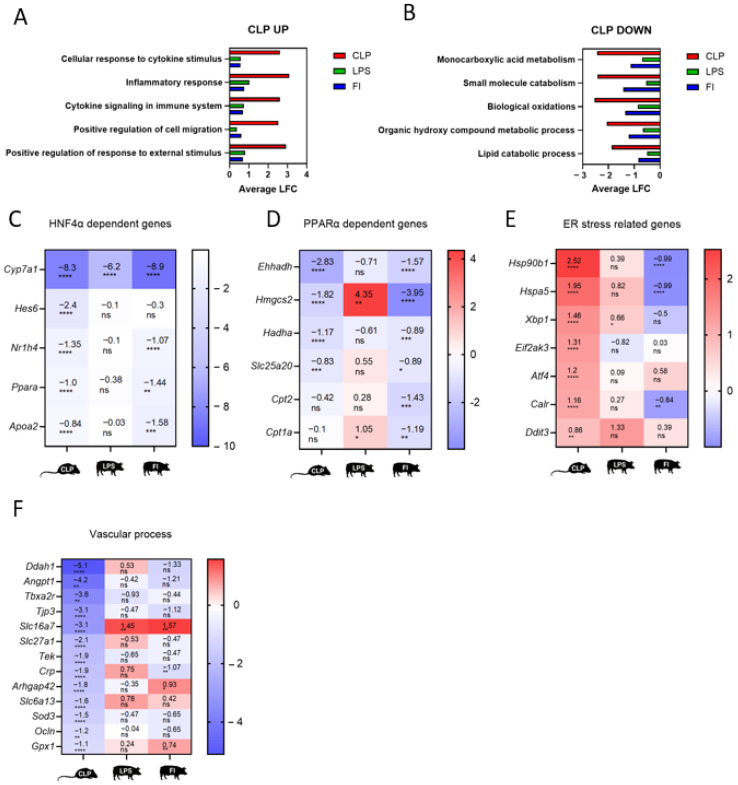
Side-by-side comparison of affected pathways based on target gene expression in the murine CLP and porcine LPS and FI model. (**A**) The top five affected pathways retrieved from the upregulated DEGs of the murine CLP model are depicted. The average LFC of genes belonging to the specified pathway is plotted for each sepsis model. (**B**) The top five affected pathways retrieved from the downregulated DEGs of the murine CLP model are depicted. The average LFC of genes belonging to the specified pathway is plotted for each sepsis model. (**C**) Heat map of selected HNF4α-dependent genes showing the LFC and significance of the specified gene compared to the respective control group for each animal model. (**D**) Heat map of selected PPARα-dependent genes showing the LFC and significance of the specified gene compared to the respective control group for each animal model. (**E**) Heat map of selected ER stress-related genes showing the LFC and significance of the specified gene compared to the respective control group for each animal model. (**F**) Heat map of selected genes related to “vascular process in circulatory system” showing the LFC and significance of the specified gene compared to the respective control group for each animal model. **** *p* ≤ 0.0001; *** *p* ≤ 0.001; ** *p* ≤ 0.01; * *p* ≤ 0.05. ns, not significant.

## Data Availability

RNA-seq data of the LPS pigs (these are newly deposited data) are deposited at the National Center for Biotechnology Information (NCBI) Gene Expression Omnibus public database (http://www.ncbi.nlm.nih.gov/geo/ accessed on 2 September 2024) under accession number GSE250039. The mouse CLP and pig FI datasets were retrieved from publicly available datasets (see Section 4).

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
