# Peer review of "Comparative Analysis of Hepatic Gene Expression Profiles in Murine and Porcine Sepsis Models"

_ijms, 2024, doi:10.3390/ijms252011079_

Round 1

Reviewer 1 Report

Comments and Suggestions for Authors

Pig is an advantagenous animal model for biomedical studies, due to its higher similarity to human in anatomy, genetics and physiology. Yet current knowledge on sepsis are largely from mice studies, with larger animals such as pigs poorly investigated. In this study, the authors compared the hepatic gene expression profiles in murine and porcine sepsis models.

They identified hundreds of genes (and relevant pathways) shared or unique between mouse and pig, or between the LPS and FI models. Specifically, they found that the inflammatory path is induced in all datasets, the ER-related pathway is induced only in mice, and the porcine FI model (but not LPS model) is more similar to the mouse CLP model. Below I listed a few comments for further improvement.

Major:

1. The authors should better explain the ultimate purpose of this study. Based on current results, it seems the major conclusion is that "the porcine FI model is more closely resembles the mouse CLP model compared to the porcine LPS model". If so, how the findings in this study can faciliatate the sepsis studies in human? Anyway, the authors are encouraged to better explain the significance of their findings.

2. The authors reached a lots of conclusions based on the motif analysis of the promoter regions (defined as -2000 bp to 50 bp from TSS) of different groups of DEGs. Promoter regions are rich with the binding motifs for many transcription factors, including many ubiquitously expressed in different tissues and conditions.  If they compared the promoters of DEGs with the promoters for other genes, the conclusions may be find. Otherwise, if they just compared the promters with shuffled backgrounds, it is very possible that many of the identified motifs are not related to sepsis. Indeed, many of the motifs they identified (such as TATA-Box, Hoxc9) are unlikely to be related to immune, stress or sepsis. The authors should provide more details about their HOMER motif analysis, and be more careful about relevant conclusions.

3. Since the two sepsis models in pigs also differ remarkably, is it possible that the differences observed between mouse and pig are just because different sepsis models are used? If so, the conclusions regarding the differences between pig and mouse will be unreliable.

Minor:

1. Apart from the overlap analysis of the DEGs between different species or different sepsis models, it is also desiable to perform correlation analysis by using the log2fold changes of all 1-to-1 orthologous genes.

2. Fig. 1 didn't reach any important conclusions. I would suggest to provide it as supplemental figure.

3. For many figures, their are too many blank areas between different sub-plots. It would be better to arrange the figures more compact.

4. The authors identified a lot of enriched motifs for down-regulated genes. I wonder if the related transcription factors are known as activators or supressors? It would be hard to explain if they are transcription activators.

5. A few typos, such as: RNA-seq is misspelled as RNA-SEQ or RNASEQ. 

Comments on the Quality of English Language

The writting is overall good, except that some paragraphs are a little lengthy.

Author Response

Reviewer 1

Comments and Suggestions for Authors

Pig is an advantagenous animal model for biomedical studies, due to its higher similarity to human in anatomy, genetics and physiology. Yet current knowledge on sepsis are largely from mice studies, with larger animals such as pigs poorly investigated. In this study, the authors compared the hepatic gene expression profiles in murine and porcine sepsis models.

They identified hundreds of genes (and relevant pathways) shared or unique between mouse and pig, or between the LPS and FI models. Specifically, they found that the inflammatory path is induced in all datasets, the ER-related pathway is induced only in mice, and the porcine FI model (but not LPS model) is more similar to the mouse CLP model. Below I listed a few comments for further improvement.

We thank the reviewer for his/her constructive suggestions.

Major:

  1. The authors should better explain the ultimate purpose of this study. Based on current results, it seems the major conclusion is that "the porcine FI model is more closely resembles the mouse CLP model compared to the porcine LPS model". If so, how the findings in this study can faciliatate the sepsis studies in human? Anyway, the authors are encouraged to better explain the significance of their findings.

Our research question is: "If the CLP (mouse) model is considered the gold standard for studying sepsis (as widely accepted in the sepsis research field), which porcine model is most suitable to follow up on findings from mice to improve translatability?" Based on our literature review, we selected the two most commonly used porcine sepsis models for comparison in this study. We have underscored the purpose of this study better in the introduction section. In the discussion section we also added more insights on how our findings could enhance translatability towards human sepsis. For example in septic mice, ER stress inhibition proved to be protective, but based on the current analysis, it seems that ER stress inhibition would not be protective in porcine, and perhaps to extension human sepsis. Furthermore, metabolic dysregulation in sepsis is now being intensively studied in the sepsis field, but based on the current data, when making the transition from mice to pigs, it is thus advisable to use the porcine FI model instead of the porcine LPS model. 

  1. The authors reached a lots of conclusions based on the motif analysis of the promoter regions (defined as -2000 bp to 50 bp from TSS) of different groups of DEGs. Promoter regions are rich with the binding motifs for many transcription factors, including many ubiquitously expressed in different tissues and conditions.  If they compared the promoters of DEGs with the promoters for other genes, the conclusions may be find. Otherwise, if they just compared the promters with shuffled backgrounds, it is very possible that many of the identified motifs are not related to sepsis. Indeed, many of the motifs they identified (such as TATA-Box, Hoxc9) are unlikely to be related to immune, stress or sepsis. The authors should provide more details about their HOMER motif analysis, and be more careful about relevant conclusions.

We have performed the HOMER motif analyses using the total genome promoter set as background. The motifs that were detected are always found as enriched compared to the background. We agree that motifs such as the TATA-Box and Hoxc9 are not (known to be directly) related to immune response or sepsis, they were found to be enriched in the set of genes compared to the background (whole genome promoter set). While we could use shuffled sequences, we believe this to negate the enrichment analysis, as the background no longer has any meaning in that case. We have added more details in the methods section about HOMER to clarify our strategy.

  1. Since the two sepsis models in pigs also differ remarkably, is it possible that the differences observed between mouse and pig are just because different sepsis models are used? If so, the conclusions regarding the differences between pig and mouse will be unreliable.

Our research question is: "If the CLP (mouse) model is considered the gold standard for studying sepsis (as widely accepted in the sepsis research field), which porcine model is most suitable to follow up on findings from mice to improve translatability?" Based on our literature review, we selected the two most commonly used porcine sepsis models for comparison in this study. These two models indeed differ remarkably at first sight, but according to the inflammatory response pathways (see original figure 5a (new figure 4A) the pig models have a very similar inflammatory response. Nonetheless, we do see very remarkable differences between the LPS and FI pig models (e.g. stronger metabolic downregulation in FI vs LPS), pointing towards the FI model better depicting the results obtained in CLP mice compared to LPS pigs. We thus advice to use the porcine FI model when studying metabolic dysfunction upon sepsis in pigs.

Minor:

  1. Apart from the overlap analysis of the DEGs between different species or different sepsis models, it is also desiable to perform correlation analysis by using the log2fold changes of all 1-to-1 orthologous genes.

We have made correlation analysis of the differentially expressed genes in CLP condition (P ≤ 0.05, LFC ≤ -1 or LFC ≥ 1) and plotted the log2fold changes of respectively FI and LPS (see new figures 2B and 3B).

  1. 1 didn't reach any important conclusions. I would suggest to provide it as supplemental figure.

This has been adapted according to the reviewer’s suggestion.

  1. For many figures, their are too many blank areas between different sub-plots. It would be better to arrange the figures more compact.

This has been adapted according to the reviewer’s suggestion.

  1. The authors identified a lot of enriched motifs for down-regulated genes. I wonder if the related transcription factors are known as activators or supressors? It would be hard to explain if they are transcription activators.

The motif enrichment method does not differentiate between activator or suppressor, it considers all upstream regulators. The interpretation for a repressor motif detected in the set of downregulated genes is simple: this repressor is predicted to be more active. For an activator TFBS motif in the downregulated genes set, the enrichment means that the activity is predicted to be reduced, either by PTM, degradation or transcriptional downregulation of the transcription factor in question.

  1. A few typos, such as: RNA-seq is misspelled as RNA-SEQ or RNASEQ. 

We have adapted this to RNA-seq in the text.

Reviewer 2 Report

Comments and Suggestions for Authors

This study compares hepatic gene expression in murine and porcine sepsis models, highlighting differences in inflammation and metabolism. Using RNA sequencing, it reveals species-specific liver responses. The detailed analysis enhances understanding of sepsis mechanisms, providing valuable insights for translating animal model findings to human sepsis research. However, there are some comments that need to be further addressed to make the manuscript better fit the hypothesis provided by the authors.

Major Comments:

1. The manuscript highlights ER stress in murine CLP models but lacks an explanation for its absence in porcine models. The authors attribute this to physiological differences, but further exploration or supporting literature is needed to clarify this discrepancy, enhancing the study's conclusions and translatability.

2. The study collects liver samples at different timepoints (24h for mice, 15-18h for pigs), impacting comparability. More justification or additional experiments aligning these timepoints are recommended to better assess gene expression and metabolic differences across models.

Minor Comments:

1. Author briefly mentions ongoing human liver biopsy studies (Line 681-682). Adding preliminary data or at least a more in-depth discussion about how these animal model findings could be translated to human sepsis would provide valuable context for readers.

2. The methods for RNA sequencing analysis, especially the bioinformatics tools used for pathway analysis, could benefit from additional details, particularly for reproducibility purposes. For example, a more thorough explanation of how HOMER and Metascape were applied would assist other researchers looking to replicate the study.

3. The pie chart depicting the usage of porcine sepsis models (Figure 1) should be reordered to present the categories in descending order, from most to least used. This numerical sequence enhances the chart’s readability, allowing readers to quickly grasp the distribution and relative prominence of each model.

Comments on the Quality of English Language

The quality of English language in the manuscript is generally good, with clear and coherent presentation of ideas. 

Author Response

Comments and Suggestions for Authors

This study compares hepatic gene expression in murine and porcine sepsis models, highlighting differences in inflammation and metabolism. Using RNA sequencing, it reveals species-specific liver responses. The detailed analysis enhances understanding of sepsis mechanisms, providing valuable insights for translating animal model findings to human sepsis research. However, there are some comments that need to be further addressed to make the manuscript better fit the hypothesis provided by the authors.

We thank the reviewer for his/her constructive suggestions.

Major Comments:

  1. The manuscript highlights ER stress in murine CLP models but lacks an explanation for its absence in porcine models. The authors attribute this to physiological differences, but further exploration or supporting literature is needed to clarify this discrepancy, enhancing the study's conclusions and translatability.

Besides RNA-seq analysis, we unfortunately do not have extra material to analyze hepatic porcine ER stress in more detail (e.g. staining or electron microscopy images). This will be subject for further research once we are able to obtain new liver samples for doing extra analysis. For the moment, we can only speculate on the explanation for absence of ER stress in the liver of porcine sepsis models. One possibility is that due to the stronger inflammatory response in murine sepsis (see original figure 5A (new figure 4A)), the ER stress is more heavily affected in mice compared to the porcine sepsis models. Indeed, it is known that inflammatory cytokines such as the pro-inflammatory cytokines IL1b, IL6 and TNF-alpha induce ER stress, which further disrupts metabolic functions, thereby causing more inflammation. This vicious cycle exacerbates the inflammatory signaling as well as metabolic deterioration. Of note, the immune system of pigs generally resemble more to humans than mice. Indeed, the lymphocyte % in blood is 75-90% in mice, 40-60% in pigs and 20-50% in humans, whereas the neutrophil % in blood is 10-25% in mice, 30-50% in pigs, and 40-75% in humans. These differences in immune composition might account for the stronger immune response to sepsis compared to pigs, and consequently more pronounced ER stress in mice. These insights have added in the discussion section.

  1. The study collects liver samples at different timepoints (24h for mice, 15-18h for pigs), impacting comparability. More justification or additional experiments aligning these timepoints are recommended to better assess gene expression and metabolic differences across models.

We acknowledge that the different timepoints are a limitation, as also noted in the discussion section. This variation arose because we collected the pig liver samples in collaboration with two different labs, which scheduled the isolation based on their own research questions. Rather than repeating the experiment to standardize the timepoints, we considered it more ethically appropriate to maximize collaboration with other groups, particularly as the isolation timepoints were nearly identical. Notably, the 15h and 18h timepoints in pigs show a comparable inflammatory response (see Fig. 4A), which justifies comparing samples with slight time variations.

Similarly, for the mouse studies, we used existing datasets from our lab to avoid unnecessary repetition of experiments with only minor differences in isolation timing, in accordance with ethical considerations. However, we also analyzed an 8h isolation timepoint in mice (dataset available in our lab), and the findings in the mice vs. pig comparison were consistent across both timepoints, reinforcing the robustness of our data. To avoid overwhelming the manuscript, we chose to present only the 24h timepoint for the mice.

See example below depicting the main findings of the manuscript. Genes depicted in the 24h CLP dataset that were respectively up- (ER stress) or downregulated (HNF4a, PPARa and vascular process related genes) show a similar trend in the 8h CLP dataset (cf. original Figure 5C-F (new Figure 4C-F)). These data point towards the robustness of the data, irrespective of the timepoint chosen. In the discussion section we shortly mention the unpublished 8h dataset and the reason why we have collected liver samples with different timepoints. (Editor's note: Please see attached word document for the figures.)

A

B

D

C

EV1: Side-by-side comparison of affected pathways based on target gene expression in the murine CLP 8h after onset versus the porcine LPS and FI model.(A) Heat map of selected HNF4α dependent genes showing the LFC and significance of the specified gene compared to the respective control group for each animal model. (B) Heat map of selected PPARα dependent genes showing the LFC and significance of the specified gene compared to the respective control group for each animal model. (C) Heat map of selected ER stress related genes showing the LFC and significance of the specified gene compared to the respective control group for each animal model. (D) Heat map of selected genes related to ‘vascular process in circulatory system’ showing the LFC and significance of the specified gene compared to the respective control group for each animal model.

Minor Comments:

  1. Author briefly mentions ongoing human liver biopsy studies (Line 681-682). Adding preliminary data or at least a more in-depth discussion about how these animal model findings could be translated to human sepsis would provide valuable context for readers.

We unfortunately do not yet have preliminary data of this clinical study. For this running clinical study we are isolating human liver biopsies for RNA-seq analysis on the one hand, and for histology on the other hand. At the moment we have liver samples stored in RNA later from 6 peritonitis patients and 7 control patients. Analysis of the human dataset will be helpful for confirming identified pathways as pointed above. Based on the current data, it remains uncertain whether inhibiting ER stress would yield significant therapeutic benefits for human sepsis patients. While preclinical studies in mice suggest a role for ER stress in the pathophysiology of sepsis, translating these findings to humans is complex due to interspecies differences in immune response and disease progression. Further studies, including clinical trials, are needed to evaluate the potential efficacy and safety of ER stress inhibition in sepsis treatment.

  1. The methods for RNA sequencing analysis, especially the bioinformatics tools used for pathway analysis, could benefit from additional details, particularly for reproducibility purposes. For example, a more thorough explanation of how HOMER and Metascape were applied would assist other researchers looking to replicate the study.

More details have been added for the HOMER analysis to the manuscript to provide these details. For the Metascape analysis, this was done with the webtool. In brief, mouse gene IDs are uploaded and processed using the default settings. We have added details and the webtool database version in the Materials and Methods.

  1. The pie chart depicting the usage of porcine sepsis models (Figure 1) should be reordered to present the categories in descending order, from most to least used. This numerical sequence enhances the chart’s readability, allowing readers to quickly grasp the distribution and relative prominence of each model.

This has been adapted according to the reviewer’s suggestion. Group ‘Other’ is put as last chart pie, since it is a collection of different uniquely used models. This Figure 1 is replaced to supplementals as suggested by the other reviewer.

Round 2

Reviewer 1 Report

Comments and Suggestions for Authors

I highly appreciate the authors' efforts during the revision. All my questions have been properly addressed.

I just have a couple minor comments to add this time:

1. Page 1 line 11: the co-first authors and corresponding author are both denoted by "#" or "*". This typo should be corrected.

2. Figure 1E&F, 2E, 3E: "p-val", "#mots", and "#backgr" are not widely used abbreviations. They should use full names unless the abbreviations are defined in advance.

Author Response

We thank the reviewer for his/her comments.

We have addressed all comments made by the reviewer;

  1. Typo has been corrected
  2. Abbreviations are defined in legend of the figures

Reviewer 2 Report

Comments and Suggestions for Authors

Re-review Comments:

1. Thank you for your detailed explanation regarding the absence of ER stress analysis in porcine sepsis models (my Major Comment 1). Your rationale, particularly the differences in inflammatory response and immune composition between mice and pigs, provides valuable context. Including this discussion in the manuscript strengthens the argument and provides a thoughtful basis for future studies. As a sepsis field researcher, I expect your Lab will have additional insights and explore this further once new liver samples are available.

2. For Major Comment 2, your rationale of the decision to avoid unnecessary repetition is reasonable and aligns with ethical research practices. Including the unpublished results of the 8h timepoint analysis in mice strengthens the robustness of your findings. The clarification added to the discussion, along with consistent trends across different timepoints, provides sufficient justification for the slight variations. However, I have a minor reminder for authors. Please be aware that varying timepoints across labs may introduce experimental bias, which could impact reproducibility in future studies. It’s worth considering this for future experimental designs.

3. I have an additional Major Comment. Upon further review, I have identified potential plagiarism issues, particularly in the Materials and Methods section, where certain portions seem closely aligned with previously published content. To avoid any ethical concerns, I recommend re-writing this section to ensure originality and to properly attribute any referenced material. It’s essential that all methods are clearly described in your own words or appropriately cited if derived from other work.

4. For Minor CommentsI have no further suggestions for revision, except regarding your ongoing human liver biopsy study. If you plan to describe this clinical proposal in the published paper, it would be advisable to include your IRB approval certification to address ethical considerations.

Author Response

We would like to thank the reviewer for his/her constructive remarks resulting in an improved version of our manuscript. All new adaptations are marked yellow in the re-submitted manuscript.

1. No further comments.

2. We have added the heatmaps comparing the 8h CLP dataset with the pig datasets in Figure S3. We acknowledge that varying timepoints across labs may introduce experimental bias and we are aware of the potential risk associated with it.

3. We have added the citations to the original paper describing the methods more clearly, and we have re-written some parts of the M&M section that had a strong similarity with these citations.

4. The reference number of the ongoing clinical study approved by the ethical committee from the University of Ghent has been added in the discussion.